

# Development and evaluation of the Parenting to Reduce Child Anxiety and Depression Scale (PaRCADS): assessment of parental concordance with guidelines for the prevention of child anxiety and depression

Wan Hua Sim[1], Anthony F. Jorm[2], Katherine A. Lawrence[1] and Marie B.H. Yap[1,2]

[1] School of Psychological Sciences and Turner Institute for Brain and Mental Health, Monash University, Melbourne, Victoria, Australia
[2] Melbourne School of Population and Global Health, University of Melbourne, Melbourne, Victoria, Australia

Corresponding author
Marie B.H. Yap,
marie.yap@monash.edu

## ABSTRACT

**Background.** Involving parents in the prevention of mental health problems in children is prudent given their fundamental role in supporting their child's development. However, few measures encapsulate the range of risk and protective factors for child anxiety and depression that parents can potentially modify. The *Parenting to Reduce Child Anxiety and Depression Scale (PaRCADS)* was developed as a criterion-referenced measure to assess parenting against a set of evidence-based parenting guidelines for the prevention of child anxiety and depressive disorders.

**Methods.** In Study 1, 355 parents of children 8–11 years old across Australia completed the PaRCADS and measures of parenting, general family functioning, child anxiety and depressive symptoms, and parent and child health-related quality of life. Their children completed measures of parenting, anxiety and depressive symptoms, and health-related quality of life. In Study 2, six subject-experts independently evaluated the PaRCADS items for item-objective congruence and item-relevance. Item analysis was conducted by examining item-total point-biserial correlation, difficulty index, *B*-index, and expert-rated content validity indices. Reliability (or dependability) was assessed by agreement coefficients for single administration. Construct validity was examined by correlational analyses with other measures.

**Results.** Four items were removed to yield a 79-item, 10-subscale PaRCADS. Reliability estimates for the subscale and total score range from .74 to .94. Convergent validity was indicated by moderate to strong correlations with other parenting and family functioning measures, and discriminant validity was supported by small to moderate correlations with a measure of parents' health-related quality of life. Higher scores on the PaRCADS were associated with fewer anxiety and depressive symptoms and better health-related quality of life in the child. PaRCADS total score was associated with parental age, parent reported child's history of mental health diagnosis and child's current mental health problem.

**Discussion.** Results showed that the PaRCADS demonstrates adequate psychometric properties that provide initial support for its use as a measure of parenting risk

and protective factors for child anxiety and depression. The scale may be used for intervention and evaluative purposes in preventive programs and research.

# INTRODUCTION

Anxiety and depressive disorders impair a child's daily functioning and are associated with long-term negative sequelae, including lower quality of life, lower academic achievement, physical ill health, poor social and coping skills, family dysfunction, unstable employment, and substance abuse (*Last, 1993*; *Kovacs & Devlin, 1998*; *Stevanovic, 2013*). In light of the disability burden contributed by anxiety and depression, a focus on preventing these internalising disorders in children is imperative (*Dozois & Dobson, 2004*; *Moffitt et al., 2007*).

An extensive body of knowledge has identified various parenting factors that influence the development and maintenance of internalising disorders in children (*Creswell et al., 2011*; *Rapee, 2012*; *Pinquart, 2017*), as well as evidence supporting the efficacy of preventive parenting programs (*Sandler et al., 2015*; *Yap et al., 2016*). Specific domains of parenting such as psychological control, overprotection and rejection, among others, have been shown to be related to child internalising disorders (*McLeod, Weisz & Wood, 2007*; *McLeod, Wood & Weisz, 2007*). Conversely, parenting factors such as encouragement of children's autonomy, warmth, involvement and monitoring are associated with lower levels of child internalising problems (*Yap & Jorm, 2015*; *Schleider & Weisz, 2016*). Moreover, some of these factors predict child internalising symptoms consistently across child development, and are potentially modifiable, e.g., child physical health, harsh discipline, and over-involved/over-protective parenting (*Bayer et al., 2011*; *Yap & Jorm, 2015*). Compared to risk factors such as genetic predisposition or socioeconomic status that are less malleable or difficult to change, parenting factors represent plausible targets for preventive interventions as they are more amenable to change, and in turn, could be effective in improving outcomes for children.

Given that the interpretations of parenting research and child outcomes are reliant on the measure used, there is a need for measures of parenting that adequately assess factors that may be relevant to the development or maintenance of specific child outcomes. Observational or researcher/clinician-rated measures to assess parenting behaviour can be complex and costly in terms of the resources required to use them reliably. Thus, the use of self-report parenting measures may be the best available alternative in settings in which direct/independent observation or child-informant data is not feasible. Measures of parenting, such as the Children's Report of Parental Behavior Inventory (*Schaefer, 1965*; *Schludermann & Schludermann, 1988*), the Parenting Scale (*Arnold et al., 1993*) and the Alabama Parenting Questionnaire (*Shelton, Frick & Wootton, 1996*) have demonstrated strong psychometric properties across countries and continue to be used today. However,
many of the existing parent-report measures conceive parenting broadly in terms of either positive (e.g., warmth) or negative (e.g., hostility) dimensions of parenting or assess only certain aspects of parenting behaviour (e.g., parent–child relationship, disciplinary practices), often with respect to child externalising behaviour or other child adjustment outcomes. One notable exception is the Multidimensional Assessment of Parenting Scale (*Parent & Forehand, 2017*) that consists of items from established measures and assesses both positive and negative dimensions of parenting that are relevant to disorders in childhood and adolescence. While global parenting measures are useful for examining the relationship between broad parenting styles or behaviour and child outcomes, use of these measures may limit the application of empirical findings to theory refinement and clinical practice (*Wood et al., 2003*). The few existing measures that assess anxiogenic or depressogenic parenting are limited in the number of parenting factors examined (e.g., Parental Overprotection Scale, *Edwards, Rapee & Kennedy, 2010*; Parenting Anxious Kids Rating Scale-Parent Report, *Flessner et al., 2017*). The absence of a measure that assesses the array of parenting risk and protective factors associated with the development of anxiety and depression in children therefore represents a gap in the current literature. Given that both risk (negative) and protective (positive) factors in parenting would presumably be of interest to researchers and clinicians, a measure that comprehensively covers parenting factors associated with childhood internalising problems, which is informed by the literature and endorsed by experts, might allow for more strategic mapping of parenting factors to target for preventive intervention.

To that end, Yap and colleagues have developed a set of *Parenting Guidelines* containing parenting strategies identified as important for preventing childhood depression and anxiety (henceforth the *Guidelines*; *Parenting Strategies Program, 2014*). These *Guidelines* were developed based on a systematic review of parental factors associated with childhood anxiety, depression, and internalising problems (*Yap & Jorm, 2015*) and a Delphi consensus study of international experts (*Yap et al., 2015*). A recent online survey study found that the *Guidelines* were well-received by parents and carers as a universal resource for the prevention of childhood depression and anxiety (*Sim et al., 2017*). Given that the recommendations in the *Guidelines* are informed by research evidence and expert consensus on parenting behaviours that can reduce the risk of child depression and anxiety, higher levels of parental concordance with the *Guidelines* are expected to have a protective effect for the child.

## Objectives and hypotheses

To assess parental concordance with recommendations in the *Guidelines*, we developed a new self-assessment parenting scale as a criterion-referenced measure. We chose to develop the scale to facilitate a criterion-referenced (CR) interpretation rather than a norm-referenced (NR) interpretation for several reasons (*Glaser, 1994*; *Shrock & Coscarelli, 2007*). In contrast to NR interpretations, in which an individual's knowledge or skills are defined relative to that of others, CR interpretations define an individual's knowledge or skills with reference to a pre-established set of specific objectives or domains of knowledge (the 'criterion'). Accordingly, a CR measure would be composed of items based on specific instructional objectives or domains of knowledge and is suited to determining whether one

is able to demonstrate a pre-defined level of knowledge or mastery of a skill (*Hambleton & Rogers, 1991*; *Glaser, 1994*). By employing a CR approach to measure parenting, we would be able to assess the knowledge and competencies a parent currently has, rather than whether a parent does as well as other parents (as per a NR measure). In addition, the areas that a parent could improve on are clearly defined by the criterion and can be targeted with specific education or intervention strategies. Changes in a parent's parenting behaviours can then be assessed against the same criterion to provide an indication of progress that is specific to the individual parent.

In this paper, we describe the development and initial validation of the *Parenting to Reduce Child Anxiety and Depression Scale (PaRCADS)*. To address the current gap in the measurement of parenting, we developed the PaRCADS to assesses the range of parenting risk and protective factors that are known to influence the development of anxiety and depression in children. Based on an assembly of modifiable parenting factors for childhood anxiety, depression and internalising problems, the PaRCADS was designed to be compatible with a transdiagnostic preventive approach. To evaluate the psychometric properties of the PaRCADS, concurrent studies were conducted with a sample of parent–child dyads and a panel of international experts. Given the similarity in some of the parenting practices tapped by the measures and the use of a common-method (i.e., parent self-report), we hypothesised that convergent validity for the PaRCADS would be demonstrated by: (a) moderate to strong correlations with other parent reports of parenting; and (b) moderate to strong correlation with parent report of general family functioning. Consistent with the multi-informant approach to assessment (*De Los Reyes et al., 2013*), we expected small correlations between child- and parent- reports of parenting. For discriminant validity, we hypothesised that the PaRCADS would have weaker correlations with parents' health-related quality of life than with other parent-report measures of parenting, given that PaRCADS was expected to assess parenting practices rather than parental health-related quality of life. In addition, based on prior work on the association between parenting and child internalising symptoms (e.g., *Yap & Jorm, 2015*; *Pinquart, 2017*), we predicted small to moderate correlations between PaRCADS and child anxiety and depressive symptoms and child health-related quality of life.

## MATERIALS & METHODS

To facilitate the evaluation of the new criterion-referenced measure of parenting, Study 1 recruited parents to complete the scale online as part of a randomised controlled trial (RCT). Study 2 was a concurrent evaluation study conducted with a panel of international experts. We integrated the findings of the two studies and assessed the reliability and validity of the final version of the PaRCADS using the sample recruited in Study 1.

### Participants and procedures
#### *Study 1 (parent–child dyad sample)*
Data were collected as part of the baseline assessment in an ongoing RCT of a web-based parenting program aimed at preventing child anxiety and depression (*Fernando et al., 2018*). Ethics approvals were obtained from the Monash University Human Research

Ethics Committee (Project numbers: CF15/4316-2015001859, 7056 and 8357) and the relevant state-specific education departments or directorates across Australia.

Participants were recruited using invitations and flyers at primary schools and community locations (e.g., libraries). Parents or carers (hereafter referred to as 'parents') of a child aged 8–11 years (inclusive), who resided in Australia, had regular access to the Internet and an email account, and were fluent in English, were eligible for the study. Only one parent and one child per family were eligible to enrol. In line with a universal preventive approach, there were no exclusion criteria in relation to participants' mental health status or history, and parents could take part in the study regardless of their child's participation status.

Interested parents were first invited to register their participation and provide informed consent for themselves and their children on the trial website. Registrations were then screened by a project manager for eligibility. Those who were eligible for the study were followed up by a member of the research team to contact the child participant at a pre-arranged time to obtain verbal assent, and then guide him or her through the online assessment over the phone as required. Clickable sound clips which dictate instructions and items were available on the website to assist children with literacy issues. Following submission of the child baseline assessment data, an automated email was triggered to send the parent participant a link to their own online baseline assessment. To minimise non-response and incomplete response rates, up to 4 reminder calls were made along with text messages or emails to encourage parents to complete their assessment.

Table 1 shows the demographic characteristics of the final sample. In total, 355 parents (89.9% mothers) registered and completed baseline assessments, which were analysed for the current paper. Parents had a mean age of 41.34 years ($SD = 5.22$). Three hundred and forty-two children (51.2% boys) also participated with their parents. Child participants had a mean age of 9.79 years ($SD = 1.05$), and 65.5% of them were in school year 3 or 4. Most parent participants had 2 or more children (85.4%), were working part-time or full-time (85.6%) and had a bachelor or higher degree (70.2%).

### Study 2 (expert sample)

The primary goal of Study 2 was to further establish the content validity of the PaRCADS by having content experts examine its items for congruence with the *Guidelines*. The use of content experts is common in establishing content validity of new tests in education and training (*Schutz, Counte & Meurer, 2007*; *Shrock & Coscarelli, 2007*; *Peirce et al., 2016*). To form an expert panel for assessing the content validity of new tests, a minimum of 3 and up to 20 experts is recommended (*Lynn, 1986*; *Gable & Wolf, 1993*).

Prospective expert participants were identified through authored publications and editorial boards of journals in the field of parenting and child mental health. Experts were invited via email to take part in the study if they had at least five years of experience in research, education and/or clinical practice in the areas of parenting and child mental health. The email invitation contained information about the purpose of the study, the response formats, timeframe and links to a downloadable and detailed information sheet and the *Guidelines*. Experts who were willing to participate in the study provided their

**Table 1** Demographics of parent participants and their children (N = 355).

|  | N | % |
|---|---|---|
| **Parent relationship to child** | | |
| Mother | 319 | 89.9 |
| Father | 31 | 8.7 |
| Step-mother | 2 | 0.6 |
| Grandmother | 1 | 0.3 |
| Guardian/Foster parent | 2 | 0.6 |
| **Parent marital status** | | |
| Single | 18 | 5.1 |
| Married/defacto | 300 | 84.5 |
| Separated or divorced | 36 | 10.1 |
| Widowed | 1 | .3 |
| **Living arrangement** | | |
| Child living with both parents in same home | 280 | 78.9 |
| Child living with parents under shared care in different homes | 30 | 8.5 |
| Child living with one parent (participant) | 41 | 11.5 |
| Child living with one parent (non-participant) | 3 | 0.8 |
| Others (e.g., foster parents) | 1 | 0.3 |
| **Parent employment status** | | |
| Unemployed | 51 | 14.4 |
| Employed part time | 190 | 53.5 |
| Employed full time | 114 | 32.1 |
| **Parent studying status** | | |
| Not studying | 294 | 82.8 |
| Studying part time | 47 | 13.2 |
| Studying full time | 14 | 3.9 |
| **Parent education level** | | |
| Year 7 to Year 12 | 20 | 5.6 |
| Trade or apprenticeship | 3 | 0.8 |
| Other TAFE/technical certificate | 36 | 10.1 |
| Diploma | 46 | 13.0 |
| Advanced diploma | 1 | 0.3 |
| Bachelor degree | 132 | 37.2 |
| Graduate diploma/certificate | 2 | 0.6 |
| Post-graduate degree | 115 | 32.4 |
| Language other than English spoken at home | 31 | 8.7 |
| Identifies self as an Aboriginal or Torres Straits Islander | 5 | 1.4 |
| **Parent history of mental health diagnosis, past or current** | | |
| Yes | 232 | 65.4 |
| No | 123 | 34.6 |

**Table 1** (*continued*)

|  | N | % |
| --- | --- | --- |
| **Child history of mental health or behavioural diagnosis, past or current** |  |  |
| Yes | 144 | 40.6 |
| No | 211 | 59.4 |

Notes.

Percentages are based on parents' report, regardless of their child's participation in the study.

informed consent via a web link hosted by the Qualtrics online survey platform. Experts could complete their review online (via Qualtrics), in soft-copy (via downloaded forms) or hard-copy (via post). The review process was estimated to take 2–3 h to complete, and experts were able to complete the task at their own time and in multiple sittings over a 6-month period.

One hundred and seventy-three invitations were sent, and fifteen experts consented to participate in the study (8.7% response rate). Six experts (five female, one male) completed the evaluation. Three experts were based in the United States, and one each in Canada, Belgium and Australia. All of them held doctoral qualifications and were involved in research activities related to parenting and child mental health. Four of them also engaged in education activities, while two were in clinical practice.

## Measures
### Study 1 (parent–child dyad sample)
#### *Parenting to Reduce Child Anxiety and Depression Scale (PaRCADS)*

*Item development and consultation with parents.* In the initial development phase, items were written by author WHS to correspond closely to the parenting strategies described in the *Guidelines* (see *Parenting Strategies Program, 2014*; for a comprehensive review of the evidence informing the inclusion of these parenting practices, please refer to *Yap & Jorm, 2015*; *Yap et al., 2015*). These items were reviewed and refined through discussions with the other authors, who have a combined expertise in child development and mental health, parenting, public health, and the development of criterion-referenced measures of parenting (*Yap & Jorm, 2015*; *Yap, Jorm & Lubman, 2015*; *Yap et al., 2016*; *Cardamone-Breen et al., 2017*). To assess the feasibility and acceptability of the items with target users, extensive consultations in the form of face-to-face workshops were carried out with reference groups of parents (see *Fernando et al., 2018* for more details). Parents' feedback was discussed by the authors before incorporating them in the refinement of the scale.

*Scoring for concordance with Guidelines.* The original parenting scale contained 83 items across 10 domains. An example item in the *Involvement in child's life* domain is "I do activities together with [child name] that [he/she] finds fun". Parents rate their parenting behaviours with reference to the target child on a five-point scale (e.g., *almost never*, *rarely*, *sometimes*, *often* and *almost always*). Questions about hypothetical situations are rated as *very unlikely*, *unlikely*, *neither likely nor unlikely*, *likely* or *very likely*. The response for each item is scored as either concordant or non-concordant with the *Guidelines* (concordant

= 1; non-concordant = 0). Ten subscale scores and a total score could be obtained by summing the corresponding item scores. For each subscale and the total scale score, the cut-off scores to indicate concordance with the *Guidelines* were deliberately set to be relatively high, to be consistent with the rationale for mastery learning (e.g., *Block, 1980*; *Wilde & Sockey, 1995*) and for the purpose of identifying areas for improvement. This method of scoring the items for concordance against a pre-determined criterion has been validated in other studies (*Yap, Jorm & Lubman, 2015*; *Cardamone-Breen et al., 2017*). The median completion time was 14 min. Items can be found in File S1.

### Child Report of Parent Behavior Inventory (CRPBI) and Psychological Control Scale (PCS)

The acceptance/rejection subscale in the latest revision of *Schaefer*'s (*1965*) CRPBI (CRPBI-30; *Schludermann & Schludermann, 1988*) and the Psychological Control Scale (PCS, *Barber, 1996*) were selected to assess children's perception of parental behaviour in some aspects of parenting that were covered in the PaRCADS. As these measures were originally validated with children 10 years and older, they were administered only to child participants 10 years or above in this study. The parent self-report version of both measures has been adapted for use by researchers and is similar to the child report version but worded to capture the parent's perspective (*Fauber et al., 1990*; *Ruiz, Roosa & Gonzales, 2002*). Items are each rated on a 3-point scale from 1 (*not like*) to 3 (*a lot like*), with higher scores indicating behaviour that is more characteristic of the parenting domain assessed. Parents reported on their own parenting in relation to the child participant. Child participants were asked to report on their participating parent only.

The 10-item Acceptance/Rejection subscale of the CRPBI (CRPBI-Acceptance) has been found to have comparable psychometric properties to its precedent versions, with acceptable internal consistency and good test-retest reliability (*Schludermann & Schludermann, 1988*; *Locke & Prinz, 2002*). An example item in the parent version is, "I am a parent who gives [him/her] a lot of care and attention", where [him/her] is customised according to the child's gender. Internal consistency reliability in this sample was high on this subscale ($\omega = .84$ for parent report and $\omega = .88$ for child report).

The PCS was originally developed as a psychological control/psychological autonomy subscale in the CRPBI. Following a factor analysis, an 8-item psychological control measure was derived as a youth-report measure of parental psychological control for children aged 10 and above, with adequate internal consistency (*Barber, 1996*). An example item in the parent report version is "I am a parent who changes the subject whenever [he/she] has something to say". In this sample, the internal consistency for the parent report version was slightly lower than the acceptable range ($\omega = .69$). Based on the child participants who completed the child-report version ($n = 144$), the omega was .76.

### General family functioning

The general functioning subscale (GF) of the Family Assessment Device (*Epstein, Baldwin & Bishop, 1983*) is designed to assess overall perceived family functioning. The GF is a summative scale containing 12 items in six dimensions: problem solving (1), communication (4), roles (2), affective responsiveness (1), affective involvement (3), and

behaviour control (1) among family members. Previous research has shown high validity and test–retest reliability for the GF subscale (*Miller et al., 1985*; *Byles et al., 1988*). Each item is rated on a 4-point scale (*strongly agree, agree, disagree* and *strongly disagree*). Higher scores indicate poorer family functioning. In this study, only parent participants completed this scale. Internal consistency was high ($\omega = .90$).

### Revised Children's Anxiety and Depression Scale (RCADS-25)

The Revised Children's Anxiety and Depression Scale short version (RCADS-25; *Ebesutani et al., 2012*) is designed to assess anxiety and depressive symptoms in children up to grade 12. The RCADS is an extension of the Spence Children's Anxiety Scale (*Spence, 1997*) by *Chorpita et al. (2000)* to include items representing DSM-defined Major Depression symptoms, negative affect and general anxiety. In both the child- and parent-report versions, responses range from 0 (*never*) to 3 (*always*), with higher scores indicating higher levels of anxiety or depressive symptoms. A total Anxiety score (15 items), a total Depression score (10 items), and a Total anxiety and depression score (25 items) can be derived from the items. The recommended clinical cut-off for each of the scores is a $T$-score of 70 or more, while a $T$-score of 65–69 would fall in the borderline range (note: $T$-scores have a mean of 50 and a standard deviation of 10). Prior research suggests that the RCADS-25 demonstrates adequate internal consistency, test-retest reliability and validity in school and clinical samples (*Ebesutani et al., 2012*; *Ebesutani et al., 2017*). The two subscale scores and the total scale score were used in this study. Internal consistency estimates for the current sample were acceptable to high on both the child- (Anxiety $\omega = .85$; Depression $\omega = .79$; Total $\omega = .89$), and parent-report (Anxiety $\omega = .85$; Depression $\omega = .82$; Total $\omega = .88$) versions.

### KIDSCREEN-27

KIDSCREEN-27 is designed to assess health-related quality of life (HRQoL) of children aged 8 to 18 years (*Ravens-Sieberer et al., 2007*). The measure contains 27 items across five dimensions: Physical well-being, Psychological well-being, Parent relations and autonomy, Peers and social support, and School environment. It has a child self-report version and a proxy-report version that can be completed by a parent or caregiver. Items are rated based on either frequency (e.g., *never, seldom, sometimes, often, always*) or intensity (e.g., *not at all, slightly, moderately, very extremely*). This measure has been validated in 13 European countries and found to have good internal consistency, test-retest reliability, and construct and cross-cultural validity (*Robitail et al., 2007*; *Ravens-Sieberer et al., 2007*). Dimension scores were used in this study. Internal consistency estimates in our sample were high in both the child- (Physical $\omega = .80$; Psychological $\omega = .88$; Parents $\omega = .86$; Peers $\omega = .86$; School $\omega = .82$) and parent-report (Physical $\omega = .80$; Psychological $\omega = .91$; Parents $\omega = .87$; Peers $\omega = .90$; School $\omega = .88$) versions.

### The Assessment of Quality of Life (AQoL-8D)

The AQoL-8D is a 35-item self-report measure of adults' HRQoL across eight dimensions: Independent living, Pain, Senses, Mental health, Happiness, Coping, Relationships, and

Self-worth (*Richardson & Iezzi, 2011*). Individual dimension scores, a Physical super-dimension score, and a Psychosocial super-dimension score can be derived. The AQoL-8D demonstrated good internal consistency, test-retest reliability, and convergent and predictive validity, with the exception of the senses dimension, which had low internal consistency reliability (*Richardson et al., 2014*). For the purpose of obtaining a general profile of parents' HRQoL, only the Physical and Psychosocial super-dimension scores were used in this study. Internal consistency estimates were high for both super-dimension scores in the current sample (Physical $\omega = .88$; Psychosocial $\omega = .96$).

### Study 2 (expert sample)

In the expert evaluation study, all participants were provided the following review materials: (1) a copy of the PaRCADS with the associated *Guidelines* and objectives each item purports to assess, and (2) a review form. For each PaRCADS item, experts were requested to provide a set of ratings and comments including two which were used in this study, namely, item-objective congruence (i.e., whether the item assesses the intended *Guidelines*), and item relevance (i.e., the extent to which the item is relevant to the *Guidelines*). Item-objective congruence was rated on a dichotomous scale (*Yes* or *No*), and item relevance was rated on a 4-point scale from 1 (*not relevant*) to 4 (*very relevant*).

### Data analytic approach

As the child report versions of the CRPBI-Acceptance subscale and the PCS were administered only to children 10 years or above, data on these parenting measures were available only for a sub-sample ($n = 144$). Results from a missing data analysis showed that about 11% of the participants in Study 1 had at least one missing value (missed 1–3 items). However, the overall proportion of missing data was less than 0.1% and the proportion of item-level missingness was less than 3% on all measures. To assess the pattern of missingness, Little's Missing Completely at Random test (MCAR) was run. Little's test suggested that item-level missing data can be reasonably assumed to be missing completely at random, $\chi^2(5941) = 5984.82$, $p = .34$. Missingness was further examined in relation to participant characteristics, such as age, gender, language spoken at home, education level and number of children, while controlling for child participation status and survey administration. At the respondent-level, younger children had more missing data ($p = .01$). At the dyad-level, parents with younger participating children ($p = .01$) had more missing data. Given that the proportion of missing data was very low, where information about imputation from the scale developers was not accessible, missing data were imputed by person mean of completed items within each subscale, to be consistent with the imputation method recommended for RCADS-25 and AQoL-8D. There were no missing data from the experts' ratings in Study 2.

Item analysis of the PaRCADS was conducted using techniques suitable for criterion-referenced measures. The item difficulty index (also known as item easiness or facility) was computed for each item to review the proportion of respondents who answered the item correctly (i.e., in this case, number of respondents who obtained a score for concordance with the *Guidelines*). The corrected point-biserial correlation ($r_{pb}$) between

an item score (scored 0 or 1) and total score was used as an item discrimination index to assess the extent to which performance on an item corresponds to overall performance in the expected direction (*Shrock & Coscarelli, 2007*). The corrected point-biserial correlation provides a more robust discrimination index as it removes the contribution of the item to the total score, and hence allows an examination of the impact of item scores on the total test (*Millman & Green, 1989*). To assess the quality of each item in distinguishing respondents who scored above the cut-point for concordance from those who scored below, the *B*-index for each item was also calculated (*Brennan, 1972*). The *B*-index was derived from the difference in item difficulty indices (see definition for item difficulty index) between respondents who scored above the cut-point and those who scored below the cut-point.

All item indices can have values ranging from −1 to +1. While there is no reference value for evaluating the *B*-index, a higher value maximally separates the masters from non-masters on a criterion-referenced test (*Brown & Hudson, 2012*). Some researchers have suggested removing items with a difficulty index or a discrimination index below or above certain values since these items do not maximise the information about differences among respondents (*Henning, 1987*; *Kline, 2013*). However, such items may be useful in their contribution to the measure or reflect critical attributes about the criterion (*Popham & Husek, 1969*). Moreover, item indices may differ across different samples of respondents or contexts (e.g., clinical *vs* community samples). Hence, where item indices fell below the recommended range, each item was inspected carefully with reference to the *Guidelines* and consideration of experts' ratings and feedback from Study 2 before a decision was made to discard or retain the item.

Descriptive statistics were used to examine the frequencies of participants' mental health history and parental concern about their child's risk of developing anxiety or depression. Independent sample *t* tests were conducted to explore possible differences in demographic characteristics and total child anxiety and depressive symptoms between children who participated in the study with their parents and those who declined to participate.

To assess the reliability or consistency of the PaRCADS in classifying parents as concordant or non-concordant with the *Guidelines*, we calculated the agreement coefficient ($p_o$) for the subscale scores and total score using the formula and approximate values provided in *Subkoviak (1988)* for single test administration. Test consistency for criterion-referenced measures is sometimes referred to as *dependability* or *consistency* rather than *reliability* so as not to be confused with approaches employed in classical test theory (*Subkoviak, 1988*; *Brown, 1990*). Where an absolute standard is chosen (as with the cut-point for concordance in the PaRCADS), and the primary interest is to measure classification consistency based on a cut-point (i.e., consistency of the PaRCADS in classifying parents as concordant or non-concordant with the *Guidelines*), the threshold loss agreement approach and the agreement coefficient were deemed to be more appropriate than other agreement indices, such as the kappa coefficient (*Berk, 1984*).

As the PaRCADS was intentionally created to be a criterion-referenced measure, with each domain containing items that represent a set of skills or knowledge necessary to be concordant with a parenting domain in the *Guidelines*, we examined the correlations

between the PaRCADS subscales scores and total score. In particular, we expected moderate to high correlations between the subscale scores and total score. Conventional factor analyses were not conducted as the items in each domain were not expected to uniformly represent a single factor in criterion-referenced measures. Construct validity was assessed by comparing the Pearson's correlation coefficients of the relationships between the PaRCADS and other measures. To explore the relationships between PaRCADS total score, participant demographics, mental health characteristics, child anxiety and depression and child HRQoL, we conducted bivariate correlation analyses and $t$-tests where appropriate.

## RESULTS

### Study 1 Parent and child characteristics

Table S1 presents the descriptive statistics for parental concerns about their child's risk of developing depression or anxiety and parent-reported mental health history. Most parents reported at least "a little" concern about their child's risk of developing depression (84.8%) or anxiety (88.5%). Nearly two-thirds of parents reported having had a prior mental health diagnosis, with 17.7% of parents experiencing current mental health problems. About two-fifths of parents reported that their children had a mental health or behavioural diagnosis in the past. Slightly more than a third of the children were reported to be experiencing current mental health or behavioural problems that had been formally diagnosed.

Non-participating children ($n = 13$) were slightly younger than children who participated with their parents ($n = 342$), $t(14.09) = -2.59$, $p = .02$. Notably, non-participating children were also reported by their parents to have higher child anxiety and depressive symptoms on the RCADS Total symptom score, $t(353) = 3.13$, $p < .01$.

### Study 2 Expert ratings

Item-objective congruence was summarised by the proportion of experts who endorsed the item as congruent with the intended *Guidelines*. Using *Lynn*'s (*1986*) guidelines, the item content validity index (I-CVI) was computed for each PaRCADS item to summarise experts' ratings on item relevance. Adopting the cut-points recommended in the literature (*Polit, Beck & Owen, 2007*; *Almanasreh, Moles & Chen, 2019*), 76 out of 83 PaRCADS items were endorsed by at least five out of six experts for item-objective congruence, and 64 items had the highest ratings from the experts for item relevance. Details of item ratings are available in File S2.

The subscale content validity index (S-CVI) was also calculated by averaging the I-CVIs for item relevance across items within each subscale. Results suggested that all subscales in the PaRCADS displayed good subscale content validity (*Davis, 1992*; *Polit, Beck & Owen, 2007*), except *Child's relationship with others* (S-CVI/Ave = .71) and *Health habits* (S-CVI/Ave = .77).

In addition to expert ratings, other item indices appropriate for criterion-referenced measures were computed for consideration as part of the item selection process.

## Psychometric properties of the PaRCADS
### Item analysis
Item statistics are available in File S2. All but nine items had difficulty indices above .30, indicating that most of the items did not suffer from floor effects. Eight items had item-difficulty indices $\geq$.90, suggesting ceiling effects for these items.

Examination of other item discrimination indices revealed that all items had positive corrected point-biserial correlations, indicating that the item scores functioned as expected in relation to the total score. Further, 54 items (65%) had corrected point-biserial correlations $\geq$.25. Items with the lowest discrimination indices (in this case, weakest corrected item-total point biserial correlations) were found in the subscale *Getting help when needed*. This was unsurprising, as items in this subscale were hypothetical questions about a parent's likelihood of taking certain actions if they noticed a change in their child's mood or behaviour, whereas most other items in the PaRCADS require parents to report on their usual parenting behaviour. An examination of the *B*-index for items in this subscale further indicates that the values were low, suggesting that the items did not strongly differentiate parents who scored at or above the cut-off score and parents who scored below the cut-off score for overall concordance with the *Guidelines*.

### Reliability analysis
Agreement coefficients ($p_o$) were used to provide estimates of the reliability/dependability of the scale. The overall measure (e.g., total score) was highly consistent ($p_o = .95$). Eight subscales had acceptable to high agreement coefficients (.75 to .92). The remaining two subscales had agreement coefficients ($p_o = .74$) just below the recommended cut-point of .75 (*Subkoviak, 1988*).

## Item reduction
After considering the inter-item correlations within subscales, corrected item-total point biserial correlations, experts' endorsement on item-objective congruence, item-relevance CVIs and feedback from experts, four items were removed due to low ratings in three or more aspects of the aforementioned. Deletion of these four items improved their corresponding subscale agreement coefficients for three subscales ($p_o$ increased by .07 for *Relationship with your child*, by .02 for *Child relationship with others*, and by .15 for *Getting help when needed*) and resulted in a reduction for one subscale ($p_o$ reduced by .02 for *Health Habits*). While there were other items with discrimination indices and expert ratings that fell outside the recommended ranges (e.g., items 2.7, 4.3), they were retained because they represented key parenting practices for the prevention of child anxiety and depression in the literature. Although items with high facility indices (i.e., too easy), such as items 1.1 and 10.1, did not maximally differentiate between respondents in this sample, they might display statistical variability in a clinical sample and were thus retained.

The agreement coefficient for the resultant total score was still very high ($p_o = .94$), indicating that reliability/dependability was high for the measure as a whole. At the subscale level, the revised agreement coefficients ranged from .74 to .90. The 79 items retained after this process were subjected to subsequent analyses. Table 2 shows the descriptive and dependability statistics for the PaRCADS revised subscale and total scale

**Table 2** Descriptive and dependability statistics for PaRCADS revised subscales and total score ($N = 355$).

| Subscale/ Domain | Highest possible score | Cut-off score for Concordance | $M$ | $SD$ | Observed minimum (%) | Observed maximum (%) | % Concordant | Agreement coefficient $p_o$ |
|---|---|---|---|---|---|---|---|---|
| Relationship with your child | 7 | 5 | 5.55 | 1.30 | 0 (0.3%) | 7 (24.2%) | 82.2 | .82 |
| Involvement in child's life | 10 | 8 | 6.70 | 1.78 | 1 (0.3%) | 10 (3.4%) | 34.7 | .74 |
| Child's relationships with others | 6 | 5 | 3.65 | 1.65 | 0 (2.5%) | 6 (16.3%) | 34.9 | .82 |
| Rules and consequences for child | 9 | 7 | 4.23 | 2.10 | 0 (1.7%) | 9 (1.7%) | 16.1 | .89 |
| Health habits | 7 | 6 | 3.10 | 1.85 | 0 (6.2%) | 7 (3.9%) | 11.5 | .90 |
| Home environment | 10 | 8 | 5.94 | 2.00 | 0 (0.3%) | 10 (1.1%) | 22.8 | .82 |
| Managing emotions | 7 | 6 | 3.96 | 1.47 | 1 (4.2%) | 7 (4.8%) | 15.5 | .85 |
| Setting goals and dealing with problems | 8 | 7 | 5.99 | 1.77 | 0 (0.3%) | 8 (23.1%) | 46.2 | .76 |
| Dealing with negative emotions | 10 | 8 | 6.55 | 1.94 | 0 (0.6%) | 10 (4.2%) | 34.4 | .78 |
| Getting help when needed | 5 | 4 | 4.52 | 0.75 | 1 (0.8%) | 5 (63.7%) | 91.3 | .89 |
| Total score | 79 | 64 | 50.18 | 10.60 | 12 (0.3%) | 74 (0.3%) | 11.3 | .94 |

scores. As shown in Table 3, there were positive correlations of varied strength between the PaRCADS subscales scores, and moderate to high correlations between each subscale score and the total score.

## Parental concordance with guidelines

The descriptive statistics for the PaRCADS revised subscale and total scale scores are presented in Table 2. Concordance rates varied across subscales. The lowest average concordance rate was observed in *Health Habits* (11.5%), while the highest was observed in *Getting help when needed* (91.3%). More than 30% of parents scored within the concordant range for six out of ten subscales. The mean PaRCADS total score was 50.18 ($SD = 10.60$).

## Associations between parental concordance and participant characteristics

Using the PaRCADS revised total score as an indicator of overall parental concordance with the *Guidelines*, we found small, negative correlations between the PaRCADS total score and parental concern about their child's risk of depression or anxiety, with higher

**Table 3  Correlations between subscale scores and total score on the revised PaRCADS ($N = 355$).**

|  | 1 | 2 | 3 | 4 | 5 | 6 | 7 | 8 | 9 | 10 |
|---|---|---|---|---|---|---|---|---|---|---|
| 1. Relationship with child | – | | | | | | | | | |
| 2. Involvement in child's life | .32** | – | | | | | | | | |
| 3. Relationships with others | .23** | .43** | – | | | | | | | |
| 4. Rules & consequences for child | .30** | .43** | .28** | – | | | | | | |
| 5. Health habits | .10* | .31** | .28** | .26** | – | | | | | |
| 6. Home environment | .29** | .31** | .24** | .40** | .22** | – | | | | |
| 7. Managing emotions | .35** | .35** | .29** | .41** | .20** | .49** | – | | | |
| 8. Setting goals and dealing with problems | .47** | .39** | .34** | .42** | .24** | .42** | .43** | – | | |
| 9. Dealing with negative emotions | .46** | .43** | .34** | .43** | .22** | .51** | .54** | .55** | – | |
| 10. Getting help when needed | .22** | .12* | .07 | .19** | .10* | .27** | .30** | .15** | .25** | – |
| *Total Score* | .57** | .67** | .57** | .69** | .49** | .69** | .68** | .72** | .77** | .34** |

Notes.

Correlations were computed after removing four items.

*$p < .05$.

**$p < .01$.

PaRCADS total scores associated with less parental concern about their child's risk (see Table 4). There were also small, negative correlations between the PaRCADS total score and parent reported child's current mental health problem ($r = -.14$, $p = .01$) and history of mental health diagnosis ($r = -.11$, $p = .03$). Of the parent's demographic characteristics, only parental age was correlated with the PaRCADS total score ($r = .10$, $p = .046$).

## Construct validity

To assess its construct validity, we examined the Pearson's correlations between the PaRCADS revised total score and the subscale scores of other parenting, family functioning and HRQoL measures.

### Convergent validity

Results of the convergent validity tests are shown in Table 5. The correlations between the PaRCADS total score and parent reports on the CRPBI-Acceptance subscale score ($r = .54$, $p < .01$) and the PCS subscale score ($r = -.42$, $p < .01$) were strong and in the predicted directions. A higher PaRCADS total score was also associated with a lower level of problematic family functioning ($r = -.52$, $p < .01$).

Parent-child agreement on closely related measures could provide another form of convergent validity. There was a small positive correlation between PaRCADS and child-report on the CRPBI-Acceptance subscale [$r(144) = .16$, $p = .03$], but not the PCS [$r(144) = -.13$, $p = .06$]. An examination of parent–child agreement on the CRPBI-Acceptance subscale and PCS showed that there were small significant correlations between parent- and child-reports on the CRPBI-Acceptance subscale and PCS ($r$s .15 and .24).

### Discriminant validity

As shown in Table 5, there was a moderate, positive correlation between the PaRCADS total score and each of the super-dimensions on the parent AQoL ($r$s .15 and .34). Using $z < -1.69$ or $z > 1.69$ for one-tailed test as the criterion, Steiger's Z test for

Sim et al. (2019), *PeerJ*, DOI 10.7717/peerj.6865

**Table 4   Correlations between PaRCADS revised total score, and parent and child characteristics ($N = 355$).**

| | Child | | | | Parent | | | | | | | |
|---|---|---|---|---|---|---|---|---|---|---|---|---|
| | Age | Gender[a] | Current mental health problem[b] | History of mental health diagnosis[b] | Age | Gender[a] | Current mental health problem[b] | History of mental health diagnosis[b] | Education level[c] | Number of children in the household | Concern about child's risk of depression[d] | Concern about child's risk of anxiety[d] |
| $r$ or $r_{pb}$ | −.02 | .04 | −.14** | −.11* | .10* | .08 | .02 | .001 | .06 | −.04 | −.18** | −.15** |

Notes.

$r$, Pearson's correlation coefficient. $r_{pb}$, Point-biserial correlation coefficient. The $r_{pb}$ was used to assess correlations between PaRCADS, gender and mental health variables.

[a]Child and parent gender was coded dichotomously where 1 = male, 2 = female.

[b]Child and parent mental health problem and diagnosis were based on parent's report and were coded dichotomously where 0 = None, 1 = presence of current/history of mental health problem/diagnosis.

[c]Parent education level was coded on a 9-point scale following the Australian Qualifications Framework with a higher score representing a higher level of education.

[d]Parental concern about child's risk of depression and anxiety was coded on a 4-point scale where a higher score indicates greater parental concern.

*$p < .05$.

**$p < .01$.

Two-tailed.

**Table 5 Convergent and discriminant validity indices.**

| | Convergent validity | | | (With child report) | | Discriminant validity | |
|---|---|---|---|---|---|---|---|
| | P-CRPBI acceptance (N = 355) | P-PCS[a] (N = 355) | GF[b] (N = 354) | C-CRPBI acceptance (N = 144) | C-PCS[a] (N = 144) | AQoL-PSD (N = 355) | AQoL-MSD (N = 355) |
| PaRCADS total score | .54** | −.42** | −.52** | .16* | −.13 | .15** | .34** |

Notes.

The indices are represented by bivariate correlations.

P-CRPBI Acceptance, Parent report of the Children's Report of Parent Behaviour Inventory-Acceptance subscale; P-PCS, Parent report of the Psychological Control Scale; GF, General functioning subscale of the Family Assessment Device completed by parents only; C-CRPBI, Children's Report of Parent Behaviour Inventory-Acceptance subscale; C-PCS, Child report of the Psychological Control Scale; AQoL, Assessment of Quality of Life 8D completed by parents only; PSD, Physical Super-Dimension; MSD, Psychosocial Super-Dimension.

[a]Higher score indicates parenting behaviour that is more characteristic of psychological control.
[b]Higher score indicates poorer family functioning.
*$p < .05$.
**$p < .01$.

correlations showed that the strength of the relationship between PaRCADS and the CRPBI-Acceptance subscale was stronger than the relationship between PaRCADS and parent AQoL (Psychosocial $z = 3.60$; Physical $z = 8.44$). Similarly, the relationship between PaRCADS and the PCS was stronger than the relationship between PaRCADS and the parent AQoL (Psychosocial $z = −9.14$; Physical $z = −7.12$). These results lend further support for the validity of the PaRCADS as a measure of parenting practices.

## Associations between parental concordance, child symptoms and child health-related quality of life

Given that the PaRCADS was intended to assess parenting practices to reduce the risk of child anxiety and depression, we expected small negative correlations between the PaRCADS total score and the RCADS subscale and total symptom scores. Further, we predicted that the PaRCADS total score will be positively correlated with child HRQoL on the KIDSCREEN.

There were small correlations in the expected directions for both parent and child reports of RCADS anxiety, depression and total scores, with lower child symptom scores being associated with higher PaRCADS total scores (see Table 6). To further explore the differences in PaRCADS total score between parents of children with borderline or clinically elevated symptoms (i.e., RCADS $T$-score >65), and those without, independent sample $t$ tests were conducted separately for parent and child report of RCADS (two subscales and total scale scores). Results showed significant group differences with small effect sizes in PaRCADS total score for depressive symptoms [$t$ (353) = 2.81, $p = .01$, $g = −.37$] and total symptoms [$t$ (353) = 2.64, $p = .01$, $g = −.37$] based on parent report only. There were no group differences based on child report of RCADS (see Table S2).

As predicted, PaRCADS total score was also moderately correlated with parent report of child HRQoL in the KIDSCREEN Parent relations and autonomy dimension, $r = .39$, $p < .01$. There were small to moderate correlations between PaRCADS and parent reports of child HRQoL in non-parenting-related dimensions such as, Physical well-being, Psychological well-being, Peers and social support, and School environment (see Table 6). Correlations between PaRCADS and child-self report of HRQoL were also significant ($r$s

**Table 6  Correlations between PaRCADS, parent- and child-report RCADS and KIDSCREEN ($N = 355$).**

| | PaRCADS total | RCADS-C anxiety[a] | RCADS-C depression[a] | RCADS-C total[a] | KY physical wellbeing[a] | KY psychological wellbeing[a] | KY parents & autonomy[b] | KY social support & peers[a] | KY school environment[a] |
|---|---|---|---|---|---|---|---|---|---|
| PaRCADS Total | – | −.12* | −.12* | −.13** | .05 | .19** | .20** | .07 | .11* |
| RCADS-P Anxiety | −.14** | .43** | .30** | .41** | −.14** | −.29** | −.15** | −.16** | −.26** |
| RCADS-P Depression | −.22** | .30** | .35** | .33** | −.21** | −.34** | −.20** | −.26** | −.29** |
| RCADS-P Total | −.19** | .42** | .36** | .42** | −.19** | −.34** | −.20** | −.22** | −.31** |
| KP Physical Wellbeing | .22** | −.17** | −.22** | −.19** | .45** | .18** | .14** | .10* | .17** |
| KP Psychological Wellbeing | .37** | −.32** | −.30** | −.33** | .23** | .38** | .21** | .22** | .35** |
| KP Parents & Autonomy[c] | .39** | −.11* | −.14** | −.13* | .08 | .24** | .30** | .12* | .19** |
| KP Social Support & Peers | .29** | −.28* | −.25** | −.28** | .19** | .33** | .24** | .32** | .30** |
| KP School Environment | .23** | −.39** | −.34** | −.40** | .20** | .34** | .19** | .29** | .53** |

Notes.

RCADS-C, Revised Children's Anxiety and Depression Scale-25 Child version; KY, KIDSCREEN-27 Child version; RCADS-P, Revised Children's Anxiety and Depression Scale-25 Parent version; KP, KIDSCREEN-27 Parent version.

[a] $n = 342$.

[b] $n = 339$.

[c] $n = 353$.

Varied sample sizes were due to unequal number of survey completers or individual subscales with missing data that were deemed as unsuitable for imputation by authors of the RCADS-25 and KIDSCREEN-27.

*$p < .05$.

**$p < .01$.

.11 to .20), except for the Physical well-being and Peers and social support dimensions. There were moderate to strong positive correlations between parent- and child- reports on the various dimensions of the KIDSCREEN. Specifically, there was a moderate, positive correlation between parent- and child-report on the Parent relations and autonomy dimension ($r = .30$, $p < .01$).

## DISCUSSION

This study supports the reliability and validity of the PaRCADS as a criterion-referenced measure that encapsulates the risk and protective parenting factors for child anxiety and depression. Systematic examination of the item statistics combined with experts' feedback resulted in the removal of four items, with the final criterion-referenced measure consisting of 79 items across 10 subscales. The agreement coefficients for the 79-item were high for the PaRCADS total scale and seven of the ten subscales. The correlations between the subscales were all significant and ranged from small to large, with higher inter-subscale correlations observed on three particular subscales: *Managing emotions*, *Setting goals and dealing with problems*, and *Dealing with negative emotions*. All subscales were positively correlated with the total scale as expected. The PaRCADS total scale score also converged with scores on two established parent-report parenting measures (i.e., the CRPBI-Acceptance subscale and PCS) and a general family functioning measure (e.g., FAD-GF). Notably, there was a strong correlation between PaRCADS and family functioning, which supports the notion of the generalisation of the quality of parenting to the family, or that parenting could act as a mediator between the family environment and child wellbeing (*Newland, 2015*). Discriminant validity for the PaRCADS was supported by a stronger relationship with other parenting measures, compared to its relationship with parent health-related quality of life (e.g., AQoL).

While the absence of a child report version of the PaRCADS limits cross-informant validation of the scale, the small correlation between child report of parenting on other parenting measure (e.g., CRPBI-Acceptance subscale) and PaRCADS at its current parent-report version offers further support of convergent validity for the PaRCADS. There was, however, no significant relationship between PaRCADS and child report on the PCS. The absence of a correlation was surprising given that parent report on the PCS was moderately associated with PaRCADS. Possible reasons for the lack of a significant association include a lack of statistical power (only a subsample of children completed the PCS) and problems in comprehension of the PCS items by some children. Future research could explore using other measures of parental psychological control that are more appropriate for children. It would also be of interest to examine the associations between the PaRCADS and other parenting measures by employing alternative multi-informant methodologies.

### Parental concordance with *guidelines*

We suggest utilising the total score on the PaRCADS for assessing the extent to which a parent's overall parenting practices align with evidence and expert consensus on parenting risk and protective factors for child anxiety and depression. Except for one subscale (*Involvement in child's life*, $p_o = .74$) where its estimate of reliability fell just below the

acceptable level, the individual subscale scores may be used if it is of interest to obtain a profile of parenting practices across different domains or to assess changes in parenting at the domain level.

In line with the notion that a carer's parenting competence can vary across different domains of social interaction (*Grusec & Davidov, 2010*), the rates of parental concordance with *Guidelines* varied widely across different domains of the PaRCADS. Based on the revised scale, *Relationship with your child* and *Getting help when needed* had the highest concordance rates (82.2% and 91.3%, respectively), while the remaining eight subscales had concordance rates ranging between 11% to 46%. Given that parents in the current sample were mostly well educated and had self-selected to participate in the study where they could receive an online preventive parenting program, and with most of their children agreeing to participate as well, it is unsurprising that parents had high scores on the two mentioned subscales. By contrast, low concordance rates were found in the *Rules and consequences* (16.1%), *Managing emotions* (15.5%) and *Health habits* (11.5%) domains. This finding may be interpreted in the context of the study sample where nearly two-thirds of the parents reported having experienced past mental health problems and 17.7% of parents were experiencing current mental health problems. However, we did not find a correlation between parents' total score on the PaRCADS and their mental health status (past or current). There were also no significant differences in the mean PaRCADS total scores between parents with or without a past ($p = .85$) or current ($p = .07$) mental health history.

In *Rules and consequences,* only 12% of parents reported involving their children in the development of family rules. This finding may be interpreted in a number of ways. Parents may think that their children lack the maturity to contract reasonable boundaries for themselves and hence do not engage their child in the process of setting rules. Alternately, parents may think that their child should be involved in developing rules but do not know how to. Notably, more than half of the parents in this study have reported use of specific rules for their child's behaviour and had reviewed or modified their rules to adapt to their child's level of maturity and responsibility. Given that the child psychopathology literature has underscored that a sense of diminished control and influence on events and outcomes in their surroundings can increase risk for anxiety disorders in children (*Chorpita & Barlow, 1998*; *Barber, 2002*), these findings suggest that it may be important to help parents develop the skills and confidence to engage their children in setting rules for self-regulation.

It is concerning that less than 16% of parents were concordant with *Guidelines* in the *Managing emotions* domain. Specifically, a large proportion of parents endorsed use of non-supportive strategies that may be perceived by the child as minimising, dismissive or controlling, which are known to undermine a child's ability to regulate their own emotions (*Fox & Calkins, 2003*). Consistent with evidence about the importance of emotion socialisation in child development (*Izard et al., 2002*; *Yap, Allen & Sheeber, 2007*), our findings point to the need for preventive efforts to support parents in developing skills in this area of parenting. Interestingly, a separate study on users of the *Guidelines* found that this parenting domain of managing their child's emotions had the highest proportion of parents reporting an attempt to change from just reading the *Guidelines* document a

month ago (*Sim et al., 2017*). This further suggests that many parents have the motivation and capacity to make changes to their emotion socialisation practices with their children.

Considering the evidence that lifestyle-related parenting behaviours influence their children's body mass index, physical activity level, dietary habits and screen use (*Jansen et al., 2013*; *Pyper, Harrington & Manson, 2016*; *Xu et al., 2018*), the very low concordance rate for *Health habits* is notable. Only 15% of parents reported that food treats such as chocolates or soft drinks were not readily accessible to their children in the home. Further, only a third of the parents reported having good health habits themselves, such as regular exercise, healthy diet and sleep hygiene. Similarly, only about a third of the parents reported having set time limits on their child's screen use. These findings highlight the need to support parents in modelling reasonable health habits and limiting the availability of high dairy/high sugar products and screen-based entertainment at home.

While it may be argued that the current cut-off scores set for parental concordance seem arbitrary, they were deliberately set to require close-to-absolute concordance with the recommendations in the *Guidelines*, given that the *Guidelines* were informed by high quality research evidence and supported by international experts. More importantly, the cut-offs for parental concordance at subscale and total scale levels were necessarily high to serve an instructional-design function in preventive programs, such that researchers or clinicians could identify areas where each parent may benefit from further development. Nonetheless, the PaRCADS can also yield continuous scores (subscale scores and total score) that may be useful in monitoring progress or in program evaluation.

## Correlates of parental concordance with *guidelines*

The small but significant correlations between parental concordance and both parent-and child-reported child anxiety and depressive symptoms provide support for the parenting *Guidelines* informing the development of the PaRCADS, and for the potential utility of the PaRCADS in preventive interventions targeting parents of primary school-aged children.

Comparing the PaRCADS total scores for parents of children above and below the RCADS borderline cut-off scores, we also found greater parental concordance among parents of children below the RCADS borderline cut-off score on the parent report. Further, the negative albeit small correlations between the PaRCADS total score and parent reported child's current or history of mental health problems suggest that there is a relationship between parental concordance with *Guidelines* and child's mental health. Although cross-sectional analyses preclude an assessment of the direction of influence, the present findings are largely consistent with the literature supporting a small but significant association between parenting and child internalising symptoms (*McLeod, Weisz & Wood, 2007*; *McLeod, Wood & Weisz, 2007*; *Van der Sluis, Van Steensel & Bögels, 2015*; *Yap & Jorm, 2015*; *Pinquart, 2017*).

The current findings of positive correlations between PaRCADS and the KIDSCREEN subscales provide preliminary evidence that parenting practices are associated with various dimensions of a child's HRQoL. This is important given burgeoning research demonstrating that children with various mental disorders are consistently rated by their parents to have lower HRQoL (e.g., *Dey, Mohler-Kuo & Landolt, 2012*). While the mechanisms of

transmission are not yet clear, these findings suggest that supporting parents in their efforts to reduce the risk of their children developing clinical anxiety and depression may be fruitful in improving their children's HRQoL.

With regard to parent's demographic and mental health characteristics, only parent's age and parental concerns with their child's risk of developing depression or anxiety were associated with parental concordance. Specifically, older parents reported higher parental concordance, and parents with higher concordance were less concerned about their child's risk of having anxiety or depression. That older parents reported themselves as more concordant with the *Guidelines* is consistent with research that found that older mothers display greater sensitivity and less hostile childrearing practices than younger mothers (*Fergusson & Woodward, 1999*; *Lewin, Mitchell & Ronzio, 2013*). Greater maturity may better prepare older parents for the stresses of parenthood and thus evolve more adaptive parenting practices than younger parents. Contrary to studies that found higher parental education level to be associated with increased knowledge about effective parenting and child development, we did not find a significant correlation between parent educational level and parental concordance. As some would contend that parent self-reported knowledge does not necessarily translate into actual practice, the mixed findings may be explained by the fact that our study examined *current practices* rather than *knowledge,* while the latter was the focus in studies that reported a correlation with education level (e.g., *Morawska, Winter & Sanders, 2009*; *Bornstein et al., 2010*). Alternatively, the relationship with parental concordance may have been attenuated by the narrow range of education level among the participants in the current sample. The finding that parents who rated themselves as more concordant with the *Guidelines* were less concerned about their child's risk of anxiety or depression is in line with a similar study that examined parental concordance with guidelines in the context of preventing adolescent depression and anxiety disorders (*Cardamone-Breen et al., 2017*). Before conclusions can be drawn about the predictive utility of the PaRCADS, further research is required to assess the relationship between parental concordance and their child's risk of developing anxiety and depression.

## Limitations, strengths, implications and future directions

Findings from this study should be considered in light of its limitations. Although only a small proportion of children declined to participate in the study with their parents, the fact that their parents rated them higher on the RCADS total symptoms than parents of participating children suggests that the current findings may not be generalisable to children with clinical levels of anxiety and depression. On the other hand, the homogeneity and composition of the participating parents—self-enrolled, highly educated—suggests that the current findings are likely generalisable to research and programs that typically attract participants with these sociodemographic characteristics, such as web-based preventive parenting programs (*Enebrink et al., 2015*; *Suárez, Byrne & Rodrigo, 2018*; *Yap et al., 2018*). These findings also need to be considered in the context of an over-representation of mothers and the age range of the children (8–11 years) in the sample.

As far as we know, the PaRCADS is the first self-assessment measure that encapsulates the range of parenting risk and protective factors in the child anxiety and depression

literature that is also supported by international experts in parenting and child mental health. Further research is needed to examine its measurement precision and to assess its reliability and validity in more diverse samples. Likewise, longitudinal data are required to establish more clearly the associations between PaRCADS scores and child anxiety and depressive symptoms. In addition, some may challenge that children's perceptions of parenting are more important, given that parent and child reports are known to be only modestly correlated (*Martin et al., 2004*; *De Los Reyes & Kazdin, 2005*). Studies that utilised multi-informant reports continue to find that parents' self-assessment of parenting behaviour explains additional variance beyond the contribution of children's report and are predictive of child outcomes (e.g., *Ratelle, Duchesne & Guay, 2017*). Future studies that employ a multi-method (e.g., clinician observation) and multi-informant (e.g., co-parent/caregiver) approach would also provide a more robust assessment of the validity of PaRCADS as a measure of parenting (*Podsakoff et al., 2003*; *De Los Reyes et al., 2013*). Another step would be to examine the utility of PaRCADS for other mental health and functioning outcomes (e.g., externalising problems, positive well-being) across different age groups (e.g., 5–7 years) and samples (e.g., clinical, cross-cultural).

## CONCLUSIONS

In this paper, we described the development and initial validation of the PaRCADS. Our findings on its psychometric properties provide preliminary support for its use as a parenting measure among parents of primary school aged children. By identifying both areas of parenting strengths and areas for further development, the PaRCADS can be used to inform preventive and intervention targets in research and clinical practice. The scale may also be used as a measure to monitor progress and evaluate change in child preventive research or programs that include a parenting component aimed at modifying risk and protective factors.

## ACKNOWLEDGEMENTS

We would like to acknowledge the schools that assisted with advertising the RCT, project manager Luwishennadige Madhawee N. Fernando, and the team of research assistants who assisted with data collection. We also acknowledge *beyondblue* for their partnership in the *Guidelines* development and dissemination. We are immensely grateful to the parents and experts for their inputs in the development and evaluation of the PaRCADS.

### Funding

This work was supported by a Monash University Faculty Strategic Grant (SGS16-0406) and an Australian Rotary Health Research Grant. Wan Hua Sim received an Australian Government Research Training Program Scholarship. Anthony F. Jorm received salary support from an NHMRC Senior Principal Research Fellowship (APP1059785). Marie BH Yap received salary support from a NHMRC Career Development Fellowship

(APP1061744). The funders had no role in study design, data collection and analysis, decision to publish, or preparation of the manuscript.

## Grant Disclosures

The following grant information was disclosed by the authors:
Monash University Faculty Strategic: SGS16-0406.
Australian Rotary Health Research Grant.
Australian Government Research Training Program Scholarship.
NHMRC Senior Principal Research Fellowship: APP1059785.
NHMRC Career Development Fellowship: APP1061744.

## Competing Interests

Anthony F. Jorm is an Academic Editor for PeerJ.

## Author Contributions

- Wan Hua Sim conceived and designed the experiments, performed the experiments, analyzed the data, contributed reagents/materials/analysis tools, prepared figures and/or tables, authored or reviewed drafts of the paper, approved the final draft, developed the scale, wrote the paper.
- Anthony F. Jorm, Katherine A. Lawrence and Marie B.H. Yap conceived and designed the experiments, authored or reviewed drafts of the paper, approved the final draft, contributed to the development and refinement of the scale.

## Human Ethics

The following information was supplied relating to ethical approvals (i.e., approving body and any reference numbers):

Ethics approvals were obtained from the Monash University Human Research Ethics Committee (Project numbers: CF15/4316-2015001859, 7056 and 8357).

## Data Availability

Sim, Wan; Yap, Marie (2018): Data for PaRCADS_Study 1 & 2. figshare. Fileset. https://doi.org/10.26180/5beba36138264.

Sim, Wan (2019): Codebook for PaRCADS_Study 1 & 2. figshare. Fileset. https://doi.org/10.26180/5c9372a2ba010.

## Supplemental Information

Supplemental information for this article can be found online at http://dx.doi.org/10.7717/peerj.6865#supplemental-information.

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
