# Peer review of "Development and evaluation of the Parenting to Reduce Child Anxiety and Depression Scale (PaRCADS): assessment of parental concordance with guidelines for the prevention of child anxiety and depression"

_PeerJ, doi:10.7717/peerj.6865_

## Round 0.1 · original submission · Major Revisions

Both reviewers offered constructive recommendations for major revisions, which I hope you will find useful if you want us to consider it for publication. My recommendation is that you follow their recommendations as much as possible.

·

Basic reporting

First, I would really like to applaud the author for this work. I believe the work presented here will be very useful to many researchers working with the prevention of mental illness in children.

In terms of basic reporting, I believe the authors use a clear and professional English throughout the manuscript, the structure conforms to standards, tables are clearly presented and informative, and two data-files are supplied.

There are a couple of things that I think should be considered before the manuscript is being accepted.

1. In the introduction (lines 88-90), the authors write “To our knowledge, no current parenting measure exists that assesses the full range of parenting risk and protective factors that are known to influence the development of anxiety and depression in children”. I agree with the authors, however, it might be worth mentioning that there are questionnaires of parental behaviors highly relevant to the development of anxiety in children (although not “full-ranging”). E.g., The parental overprotection scale (e.g., Clarke, Cooper & Creswell, 2013) and Rearing behaviors questionnaire (e.g., Bögels & Melick, 2004). I suggest the authors mention these measures and briefly discuss in what way the paRCADS add to these existing measures.

2. The hypotheses presented in the Introduction (lines 130-135) seems somewhat arbitrary. I suggest that the authors present more background (or rational) of the expected directions and strengths presented here.

Experimental design

I believe the research is within the scope of the journal, that the research question is adequate, and that the manuscript involves a methodologically comprehensive approach.

There is one thing that could be improved, which I believe would strengthen the interpretation of the results later on:

1. Regarding recruiting participants (lines 150-158), I think it could be more clearly described what population the intervention (web-based parental program) was targeted to. This is important, as later in the manuscript it is obvious that this is not a “normal” population. For example, about 2/3 of parents had previous mental health disorders. Already at this stage in the manuscript it would be clarifying to describe the sample better.

Validity of the findings

From what I can see, data seems to be robust and adequately handled. And, I believe conclusions are generally well stated.

However, there are a few things that I would like to comment, that I think may be important to consider before accepting the manuscript.

1. Two things important to consider when exploring convergent (and discriminant) validity of a parent self-report measure is (1) the common-method bias (or common method variance) and (2) the general discrepancies of parent-child ratings. Common method variance means that the correlations between different parent-measures may be inflated. And contrary, the general discrepancies of parent-child ratings mean that the correlations between a parent-rated and child-rated questionnaire is expected to be weak. These commonly reported phenomenon could be good to highlight in order to understand why the authors specified “We hypothesised that convergent validity … would be demonstrated by: a) moderate to strong correlations with other parent reports of parenting; b) small correlations with child reports of parenting”. Further, highlighting these phenomenon may also help the reader to interpret the result section, e.g., why the child-rated parenting only showed a small correlation to paRCADS.

2. I agree that a weaker correlation between paRCADS and parental QoL compared to paRCADS and other measures of parenting support discriminant validity (i.e., evidens that the questionnaire measure aspects of parenting behaviors rather than parental QoL). However, also interesting is another level of discriminant validity, namely, whether the paRCADS really measures parenting behaviors related to risk/protective factors of anxiety/depression (as stated) or risk/protective factors of mental health in general.

Looking at the numbers (in Table 7), it seems that there is a stronger (positive) correlation between paRCADS and psychological well-being compared to the correlation (negative) between paRCADS and child anxiety/depression. This is true for both parent-ratings and child-ratings.

Perhaps the authors could discuss these findings. Could the paRCADS really be viewed as measuring parenting related to depression/anxiety? Or should it be defined to measure parenting related to mental health more broadly?

3. In the discussion section, the authors discuss the percentage of parents that were in concordance of the criteria. I believe this discussion would benefit from more clearly considering the specific sample examined in this study. Basically, it could be more clearly stated that this discussion concerns a specific group of parents (i.e., 2/3 with a prior mental disorder).

For example, the authors are concerned with the small number of parents that used adequate parenting regarding “managing emotions”. However, given this group of parents, this may not be so surprising, also, this result may not be replicated in a population-based sample.

Additional comments

Some minor issues:

1. They study involves a rather small age-range (8-11 years), perhaps add to limitations. Or propose to explore wider age-range in future research.

2. Define T-score (line 285), it may not be common knowledge what a T-score is (i.e., Mean 50, SD=10).

3. This sentence (line 337-338) is ambiguous “Little’s MCAR test suggested that item-level missing data can be reasonably assumed to be missing completely at random…”. Perhaps provide statistic details.

Reviewer 2 ·

Basic reporting

1) I found this paper used the clear and unambiguous English throughout the paper. However I think that too much detailed description/information (which I believe is best for replication) in Materials and Methods & Results sections makes it difficult for readers to follow: for example, the specific values from the supplementary tables (e.g., line 544-546) authors mentioned in results section. Please find the excessively described writings and rephrase them in short throughout the manuscript.
2) I think it is the same name/concept as the number of questions from Table 2 and the highest possible score from Table 3. Furthermore, the numbers of questions which are different in Table 2 and Table 3 makes readers feel confused how many items the subscale of PaRCADS consists of. I suggest you would delete Table 2 and add the example items from Table 2 into the text.
3) I suppose if the Table 5 were presented as landscape view not as portrait view like Table 6, it would look nicer.
4) Raw data has not been provided in the supplementary files. I suggest that you upload the raw data or explain the reason if you can’t.
5) I can say authors conform the PeerJ standards very well in References section except the line 767: italicize ‘Educational and’ because it is the title of the Journal.

Experimental design

1) I find this study would contribute very necessary guidelines for the prevention and intervention of the mental health science based on the methodological soundness. However, authors mentioned relatively unfamiliar names of statistical methods such as point-biserial correlation, Little’s MCAR test, B-index, classification-consistency estimate, etc., so that some readers might not know what they are. Please explain the concept of them in short.
2) Study 2 Expert ratings seems important for item selection using item content validity index (line 421-433). However, the 79 items were selected as the final total items though the study 2 reported that 76 items were endorsed by most experts. In addition, the study 2 repeatedly mentioned the file S2 and specific values again so that it might make readers puzzled to understand. Please find alternative ways to describe them (e.g., summarize them briefly not in separate section).

Validity of the findings

I understand the internalizing disorders are related to parenting factors such as overprotection and rejection according to the cited articles published by McLeod, Weisz & Wood (2007) and by McLeod, Wood & Weisz (2007). However, parental rejection and overprotection is associated with externalizing problems (e.g., Hale III, W. W., Van Der Valk, I., Engels, R., & Meeus, W. (2005). Does perceived parental rejection make adolescents sad and mad? The association of perceived parental rejection with adolescent depression and aggression. Journal of Adolescent Health, 36(6), 466-474; Buschgens, C. J., Van Aken, M. A., Swinkels, S. H., Ormel, J., Verhulst, F. C., & Buitelaar, J. K. (2010). Externalizing behaviors in preadolescents: familial risk to externalizing behaviors and perceived parenting styles. European child & adolescent psychiatry, 19(7), 567-575.). Considering that the latter article published by Buschgens and colleagues (2010) are utilizing longitudinal data, it is very helpful to understand the risk and protective parenting factors. Therefore I suggest you’d better add the possibility of the study that PaRCADS is good to deal with both internalizing and externalizing disorders.

Additional comments

I agree that the PaRCADS is valuable instrument to evaluate and intervene the parenting in research and clinical practices. If the title of the manuscript is changed into validation or development of PaRCADS: ~, it would be much easier for readers to grasp the contents of the paper.

---

## Round 0.2 · accepted · Accept

You have been careful and responsive to the prior reviews, which has improved the clarity of the manuscript in this important area of research.

#